# The emergence of moral alignment within human groups is facilitated by interbrain synchrony
Aial Sobeh [1] ✉ & Simone Shamay-Tsoory [1,2]

Humans tend to align their behaviors and beliefs with their group peers. Establishing alignment between group members is crucial for group unity, yet the mechanisms underlying its emergence are under-explored. Here we examined the extent to which the brains of group members synchronize during deliberation on moral issues, and how interbrain synchrony supports alignment in their moral beliefs. We scanned 200 participants, who were divided into groups of four, using functional Near-Infrared Spectroscopy (fNIRS) during discussions on moral dilemmas. Behavioral results show that following group deliberations, members aligned their beliefs by adjusting their private beliefs towards the collective sentiment. Critically, neuroimaging results reveal that increased interbrain synchrony in the left inferior frontal gyrus (IFG) between group members predicts the degree of alignment post-deliberation. These findings indicate that the human tendency to align with group members extends to moral beliefs and reveal that regions related to mirroring and semantic sequence processing work across brains in coordination, to promote shared moral beliefs.

From synchronizing footsteps while walking to adhering to peer-established norms, individuals frequently adjust their behavior to align with that of others[1,2]. Social alignment, broadly defined as the process by which individuals adjust their behaviors, beliefs, or emotions to match with those of others, plays a pivotal role in fostering connectedness and group cohesion within human communities[3–6]. A specific form of social alignment, which we refer to as moral alignment, occurs when individuals adjust their moral beliefs to conform to perceived group norms[7,8]. This form of alignment minimizes deviations among group members from established group norms, helps maintaining cohesion in moral conventions, and facilitates collective decision-making on ethical dilemmas[9,10]. Aligning group members around common moral principles is essential for guiding, constraining, and coordinating their collective actions when addressing moral issues facing the group[11–13].

While previous research on moral beliefs change provides evidence for the flexibility of moral views and the tendency of individuals to align with perceived majority opinions[7,8], these studies largely overlook the interactive processes through which humans arrive at moral agreements. Specifically, little is known about how social deliberation—a collaborative and dynamic form of interaction—drives moral agreement within groups faced with morally ambiguous dilemmas. Groups constantly encounter new challenges marked by moral ambiguity, leading to a lack of moral agreement among members and a lack of established norms for them to follow[14–17]. Disagreement in moral beliefs between individuals can stem from differences in the

moral reasoning guiding personal interpretations and responses to moral challenges[18]. A key process through which these disagreements are managed is consensus-oriented deliberation—a dynamic social process through which individuals exchange arguments and potentially converge on shared moral judgments[19,20]. We rely on this kind of discussions to address many morally controversial issues that demand reaching a consensus decision least likely to face resistance from group members[19,20]. In some cases, these discussions can offer an extensive interchange of outlooks and reasons, ultimately aligning beliefs around a reached consensus[21,22], while in other cases they can fail or even exacerbate divisions and polarization. Here we investigate the condition under which *moral alignment*—defined as alignment in how group members construe and judge moral issues—evolves through deliberation within groups facing novel and morally ambiguous issues.

To examine when and how the deliberation process succeeds in aligning divergent beliefs, we focus on the interbrain neural processes that take place during social interaction[23–25]. Aligning divergent beliefs through discussions is an interactive process that involves multiple communicating individuals attempting to influence, understand, and learn from each other[25]. Therefore, examining the interbrain relationships during communication can illuminate when deliberation is likely to align beliefs among group members[1]. Yet, prior neuroimaging research on social alignment has typically examined brain activity in non-interactive tasks that restricted communication. For instance, participants were often tasked with

[1]Social and Affective Neuroscience Lab, Department of Psychology, University of Haifa, Haifa, Israel. [2]The Integrated Brain and Behavior Research Center (IBBRC), Haifa, Israel. ✉e-mail: aealsobh123@gmail.com

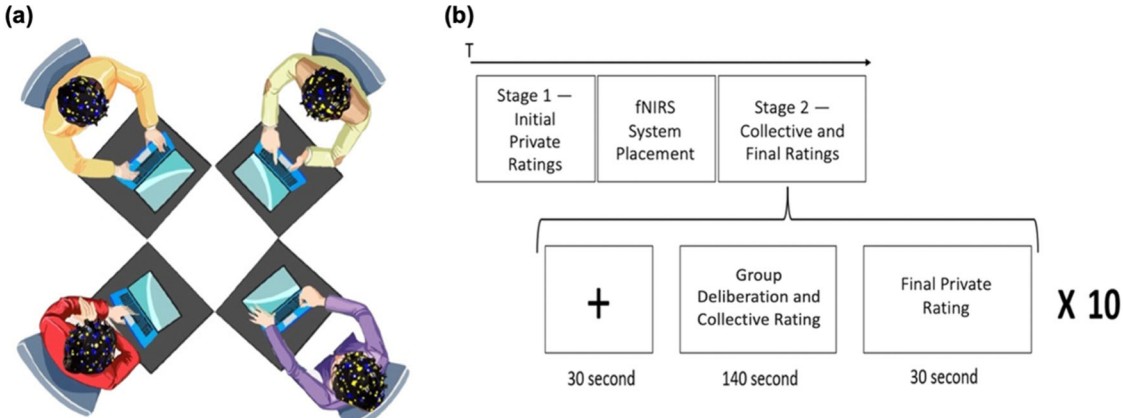

**Fig. 1 | Seating arrangements and task flow. a** Seating arrangements: the four group members are seated at separate tables facing each other and their personal computers. **b** Moral judgment paradigm timeline (described in detail in the "Methods" section).

expressing their preferences (e.g., rating the attractiveness of an item or agreeing with a statement) before being exposed to an experimentally manipulated group influence, such as a majority opinion differing from their own[26–29]. The researchers would then investigate the neural activity associated with changes in personal attitudes toward the group's preference (a measure of alignment) following the exposure to social information. However, by restricting verbal expression and direct social interaction, these studies offer a limited view of the dynamic processes through which alignment emerges during social deliberation. A full account of the process through which group members align their thoughts through deliberation requires a hyperscanning approach that captures how multiple brains communicate to influence and infer each other's mental states[2,23,30].

Recent hyperscanning studies have reported that the neural activity of interacting individuals becomes coupled during social interactions[31,32]. Interbrain synchrony represents coherence in neural activity time series of two brains, capturing the interdependent and concurrent activation patterns across those brains[23], and it is commonly observed during joint social tasks that require communication and coordination[31,32]. The observation-execution (mirror) neuron system[33] has emerged as a core system that shows interbrain synchrony during social interaction. Several hyperscanning studies involving communicative tasks have found increased interbrain synchrony specifically in the inferior frontal gyrus (IFG) with an important distinction between interbrain synchrony in the left IFG and the right IFG[34,35]. While the right IFG seems to be coupled during non-verbal coordination such as mutual gaze, joint attention, and humming[36–38], the left IFG is mainly coupled during verbal communication tasks. The left IFG was found to be coupled between interlocutors during face-to-face communication, higher in dialog compared to monologue, and higher in cooperative compared to solo tasks[35,39]. Additionally, left IFG interbrain synchrony predicts better coordination and stronger feelings of rapport[35,40], and predicts social learning outcome[41,42]. Importantly, interbrain synchrony in the left IFG between a speaker and a listener was observed during storytelling and narrative conveying, with increased synchrony predicting improved comprehension[43,44]. Based on these findings, interbrain synchrony in the IFG was suggested to reflect coupling in the same neural circuitry underlying both production/output and observation/input systems of interacting partners, allowing them to model, understand and adapt to each other[43,44].

The current study investigated if interbrain synchrony during social deliberation can predict the emergence of moral alignment. Building on the aforementioned findings, we propose that interbrain synchrony in the left IFG facilitates information exchange between deliberating partners and enhances their ability to align their beliefs. Accordingly, we expected that the extent of interbrain synchrony in the left IFG should vary between deliberations and can distinguish those that lead to moral alignment from those that do not.

To test this hypothesis, we measured the simultaneous brain activity of group members as they engaged in a moral judgment task. Previous research highlights the critical role of group size in shaping dynamics and synchrony-related phenomena[45]. In groups, synchrony fosters complex communication networks and subgroup dynamics that are absent in dyadic interactions. Additionally, studies on small-group behavior in mammals show that groups of four exhibit movement patterns that mirror the underlying social interaction maps and dynamics of the group[46]. Building on these findings, each group consisted of four members who were simultaneously scanned with functional near-infrared spectroscopy (fNIRS) (Fig. 1a). The paradigm involved presenting participants with ten trolley-type moral dilemmas[47], wherein a protagonist chooses between two harmful outcomes, involving, for example, a decision whether to actively intervene to sacrifice one life in order to save multiple lives (a "utilitarian" decision), or to uphold the principle of doing no active harm and refrain from intervening, risking more deaths (a "deontological" decision). Participants rated the morality of the described action in each dilemma three times: first privately, then as a group (consensus decision) after deliberation, and then again privately (Fig. 1b). This experimental design allowed us to derive a moral alignment measure, computed as the degree to which group members adjusted their initial private ratings in the direction of the group consensus when they provided their final private ratings. Our first analysis tested if group members whose moral judgments were initially not aligned would align their judgments following deliberations by converging around the emergent group's consensus. Importantly, we hypothesized that higher left IFG interbrain synchrony between group members during deliberation would predict higher levels of post-deliberation alignment of privately held moral views toward the group consensus. The results support our hypothesis.

## Results
### Group deliberation elicits moral alignment
To test whether group members converge their personal ratings following deliberations, we examined whether the variance in their post-deliberation private ratings (final ratings) in any given dilemma was lower than in their pre-deliberation private ratings. We predicted that variance would be lower in post-deliberation ratings, indicating that members aligned their private ratings following deliberation. Using Hierarchical Linear Modeling (HLM), we modeled the relationship between "variance" as a DV and "condition" (pre-deliberation compared to post-deliberation) as a dummy-coded binary IDV, with group as a random factor and allowing the intercept to vary by group (random intercept model). Pre-deliberation was dummy-coded as zero and post-deliberation was coded as one. As predicted, the model found a significant decrease of variance post-deliberation compared to pre-deliberation, as indicated by the significant negative slope ($p < 0.001$; see Table 1), showing that group members indeed converged their private

**Table 1 | The effect of condition (pre-deliberation compared to post-deliberation) on variance in private ratings**

| Model | Term | Mean (SE; 95% CI) | $t$ (df) | $P$ |
|---|---|---|---|---|
| Random intercept model | Intercept | 5.924 (0.167; 5.591, 6.257) | 35.33 (68.16) | <2e-16 |
| | Slope | −2.501 (0.138; −2.772, −2.229) | −18.06 (893) | <2e-16 |

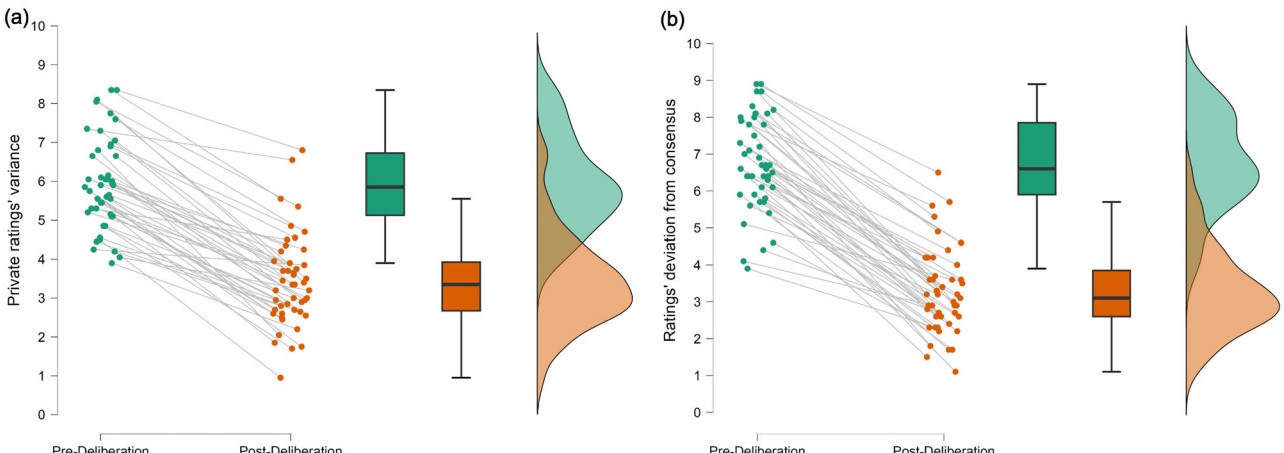

**Fig. 2 | Deliberations induce moral alignment. a** Variance in private ratings is significantly lower in final ratings (post-deliberation) than in initial ratings (pre-deliberation). Each line represents the slope of a single group ($n = 47$). **b** The deviation of private ratings from the consensus is significantly lower in final ratings (post-deliberation) than in initial ratings (pre-deliberation). Each line represents the slope of a single group ($n = 47$). The variability of data is represented in error bars, where the box captures the middle 50% of the data, with its upper and lower edges corresponding to the third and first quartiles, respectively, and the line inside indicating the median. The whiskers extend to 1.5 times the IQR, illustrating the overall spread of the data.

**Table 2 | The effect of condition (pre-deliberation compared to post-deliberation) on deviation of private ratings from consensus**

| Model | Term | Mean (SE; 95% CI) | $t$ (df) | $P$ |
|---|---|---|---|---|
| Random intercept model | Intercept | 6.736 (0.173; 6.392, 7.08) | 38.92 (73.28) | <2e-16 |
| | Slope | −3.459 (0.155; −3.763, −3.155) | −22.32 (893) | <2e-16 |

ratings following deliberation (Fig. 2a). The variance explained by the model ($R^2$) was 0.351, and the effect size ($f^2$)—calculated as ($R^2/1−R^2$)—was 0.540, corresponding a large effect size[48]. Furthermore, the intercept in our model, whose value corresponds to the average variance in pre-deliberation ratings, was significantly higher than zero (see Table 1), indicating that the initial state of individual ratings is characterized by a lack of alignment. We compared our random intercept model to an unconditional means (null) model and a random slope model and found that the random intercept model explains the variance significantly better than the null model, whereas the random intercept model and random slope models do not differ significantly (Supplementary Table 1 details models comparison results).

We further tested whether group members aligned their ratings with the reached consensus following deliberations, we examined whether the deviation of their post-deliberation private ratings from the consensus in any given dilemma was lower than in their pre-deliberation private ratings. The deviation was measured as the absolute difference between a personal rating and the consensus, which is then averaged over group members to obtain one value per group per dilemma. We predicted that deviation would be lower post-deliberation than pre-deliberation, indicating that participants indeed aligned their private ratings with the consensus. Using HLM, we modeled the relationship between "deviation" as DV and "condition" (pre-deliberation compared to post-deliberation) as a dummy-coded binary IDV, with group as a random factor and allowing the intercept to vary by group (random intercept model). Pre-deliberation was dummy-coded as zero and post-deliberation was coded as one. As predicted, we found significantly lower deviation in ratings post-deliberation compared to pre-

deliberation, as showed by the significant negative slope ($p < 0.001$; Table 2), indicating that group members aligned their ratings with the reached consensus following deliberations (Fig. 2b). The variance explained by the model ($R^2$) was 0.405, and the effect size ($f^2$)—calculated as ($R^2/1−R^2$)—was 0.680, corresponding a large effect size[48]. We compared our random intercept model to an unconditional means (null) model and a random slope model and found that the random intercept model explains the variance significantly better than the null model, whereas the random intercept model and random slope models do not differ significantly (Supplementary Table 2 details models comparison results).

Together, behavioral analyses demonstrate that group members faced with morally ambiguous dilemmas often disagree on the right course of action, yet following deliberations, they tend to align their private beliefs by adopting the collective sentiment (i.e., the consensus ratings).

## Interbrain synchrony is higher in real groups than in pseudo groups

To examine if during deliberation groups exhibit an increase in interbrain synchrony, we compared levels of interbrain synchrony in real groups to pseudo groups. Using a linear regression model, we modeled the relationship between "group-level interbrain synchrony" as DV and "group type" as a dummy-coded IDV (pseudo groups were dummy-coded as zero and real groups were coded as one), and with region of interest (ROI) pairing as a second categorical IDV with an added interaction term.

The model found a significant increase of group-level interbrain synchrony in real groups compared to pseudo groups, as indicated by the

**Fig. 3 | Assessing the effect of group type (real compared to pseudo groups) on group-level interbrain synchrony. a** Data for HHb signal. **b** Data for O2Hb signal. Each colored line represents the slope of a different interbrain edge. The *y*-axis represents interbrain synchrony, measured as the wavelet transform coherence of neural signals between participants' left inferior frontal gyri (IFGs) during deliberation. Higher values indicate stronger neural synchrony between group members.

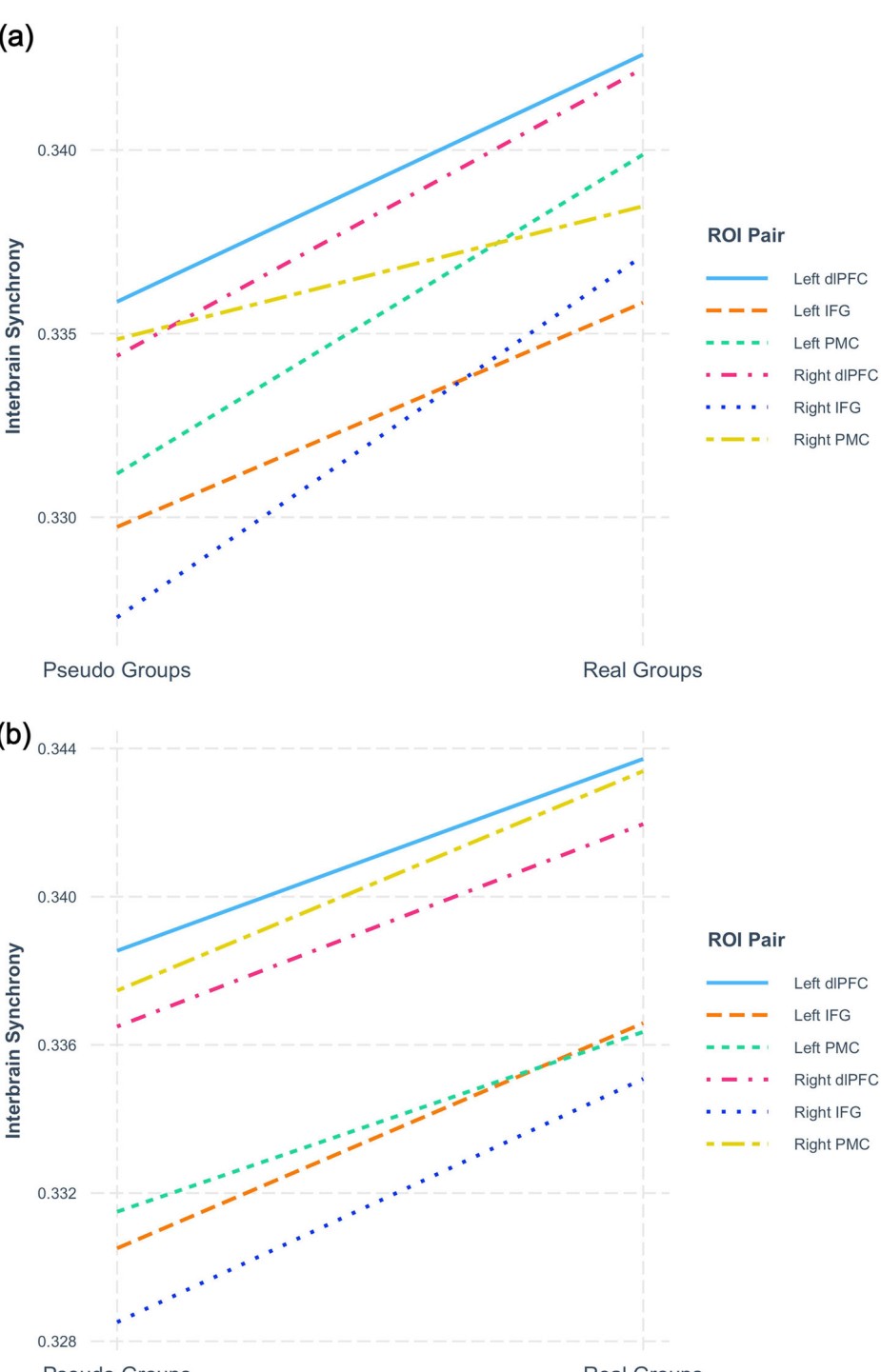

significant positive slope ($p < 0.001$; Fig. 3a), (see Table 3). Furthermore, there was no significant interaction effect between the two IDVs group types and interbrain edge (see Table 3). This result suggests that the significant increase in interbrain synchrony in real groups compared to pseudo groups does not vary significantly between the different interbrain ROI pairings. Follow-up contrast analysis (Supplementary Table 3), while applying the Tukey Multiple Comparison test to adjust for or the multiple testing, shows that interbrain synchrony is significantly higher in real groups compared to pseudo groups for all interbrain ROI pairings except for the right premotor cortex (see Supplementary Table 4 for descriptive statistics).

We replicated these results using O2Hb signal data, demonstrating significantly higher interbrain synchrony in real groups than in pseudo

groups, overall ROI pairings ($p < 0.001$; Fig. 3b) (Supplementary Table 5), and no significant interaction effect between the two IDVs group types and ROI pairings.

### Left IFG interbrain synchrony predicts moral alignment
To test the main hypothesis that increased interbrain synchrony between group members in the left IFG predicts an increase in group moral alignment, we modeled the relationship between the two variables using two HLMs, with group as a random factor. In the first model, only the intercept was allowed to vary by group, whereas in the second model, both the intercept and the slope were allowed to vary by group. In the first model, increased left-IFG interbrain synchrony during deliberation was found to

predict group moral alignment ($F$ (363.7) = 3.92, $p < 0.048$) (Fig. 4; Table 4—random intercept model). The variance explained by our model ($R^2$) was 0.055, and the effect size ($f^2$)—calculated as ($R^2/1-R^2$)—was 0.058, corresponding to a small effect size[48]. To ensure that this effect was specific to the left IFG, we conducted an exploratory analysis of all other potential interbrain ROI pairings, applying Bonferroni correction to control for multiple comparisons. This analysis found no effect in any other areas (see Supplementary Table 6 for a detailed statistical summary for models of all interbrain ROI pairings).

We conducted the same analysis for O2Hb signal data, but the results did not replicate the HHb signal results. We used a hierarchical linear model with group as random factor and allowed the intercept to vary by group. This model found no effect of interbrain synchrony on moral alignment for any of the ROI pairings after applying Bonferroni correction for multiple comparisons (see Supplementary Table 7), indicating that the effect is specific to the HHb signal.

To further examine the relationship between (HHb) left IFG interbrain synchrony and group moral alignment, we modeled the relationship using a second modified hierarchical linear model with both random intercept and random slope coefficients, thereby allowing both the intercept and the slope to vary by group. As in the random intercept model, we found that increased left IFG interbrain synchrony during deliberation predicted moral alignment ($F$ (385.7) = 3.96, $p < 0.047$) (Table 4—random slope model). Comparing the random intercept model to the null model revealed a significant difference in the goodness of fit, whereas comparing it to the random slope model revealed no significant difference (Table 5), indicating that allowing the slope to vary by group does not improve the model fit.

Nonetheless, allowing the slope to vary by group yielded a unique slope for each group, which enabled us to conduct a follow-up exploratory analysis to examine whether any group-level factors influenced estimated slope values for the different groups. First, we examined whether the slope value was dependent on group composition (mixed gender, non-mixed) by conducting two-sample student $t$-tests, with group composition as the independent variable and slope as the dependent variable. We found no significant difference between the mixed and the non-mixed groups ($t$ (45) = −0.568, $p = 0.57$). Next, we examined whether the slopes were dependent on group dominance structure by running Spearman's correlation test. Group dominance structure (see supplementary method section) captures the dominance equality between group members by measuring the variance of dominance scores between group members (i.e., a "completely egalitarian" group would have zero variance). The results of the Spearman's correlation test indicated no significant correlation between the two variables, ($r$ (45) = 0.246, $p = 0.094$). The finding that the strength of the association (estimated slope) between interbrain synchrony and group moral alignment is not affected by group-level factors, such as group gender composition or dominance structure, points to the relative robustness of the association.

## Discussion

Despite the ubiquity of moral norms and collective moral judgments, very little empirical investigation has been dedicated to exploring the mechanisms underlying the emergence of moral alignment in groups. Our findings demonstrate that moral alignment emerges through consensus-oriented

**Table 3 | ANOVA results for the linear regression model showing the effect group type (real groups compared to pseudo groups) and ROI pairing on interbrain synchrony**

| Source | DF | SS | MS | *F* value | *p* value |
|---|---|---|---|---|---|
| Group Type | 1 | 0.0074 | 0.0074 | 76.277 | <0.0001 |
| ROI pairing | 5 | 0.0041 | 0.0008 | 8.42 | <0.0001 |
| Interaction (group type × ROI pairing) | 5 | 0.0005 | 0.0001 | 1.1906 | =0.3123 |

The results were obtained using the *anova()* function in R software provided with the linear regression model as an argument. Predicted variable is interbrain synchrony (HHb signal).

**Table 4 | HLM model showing the effect of left IFG interbrain synchrony on group moral alignment**

| Model | Term | Mean (SE; 95% CI) | *t* (df) | *P* |
|---|---|---|---|---|
| Random intercept model | Intercept | −0.022 (1.77; −3.5, 3.45) | −0.013 (357.7) | 0.99 |
| | Slope | 10.47 (5.29; 0.12, 20.83) | 1.98 (363.7) | 0.048 |
| Random slope model | Intercept | −0.017 (1.76; −3.46, 3.43) | −0.01 (396.45) | 0.99 |
| | Slope | 10.45 (5.25; 0.17, 20.75) | 2.0 (385.7) | 0.0472 |

**Fig. 4 | The effect of left IFG interbrain synchrony on group moral alignment.** The *x*-axis represents interbrain synchrony, measured as the wavelet transform coherence of neural signals between participants' left inferior frontal gyri (IFGs) during deliberation. Higher values (range: from 0 to 1) indicate stronger interbrain synchrony between group members. Each data point represents a single group's interbrain synchrony and moral alignment observation. The blue regression line represents the fixed effect, showing that increased left IFG synchrony predicts greater moral alignment within groups. Specifically, a one-unit increase in interbrain synchrony corresponds to a 10.47-unit increase in group moral alignment ($p < 0.05$). The gray regression lines represent the estimated slopes and random intercepts for each group.

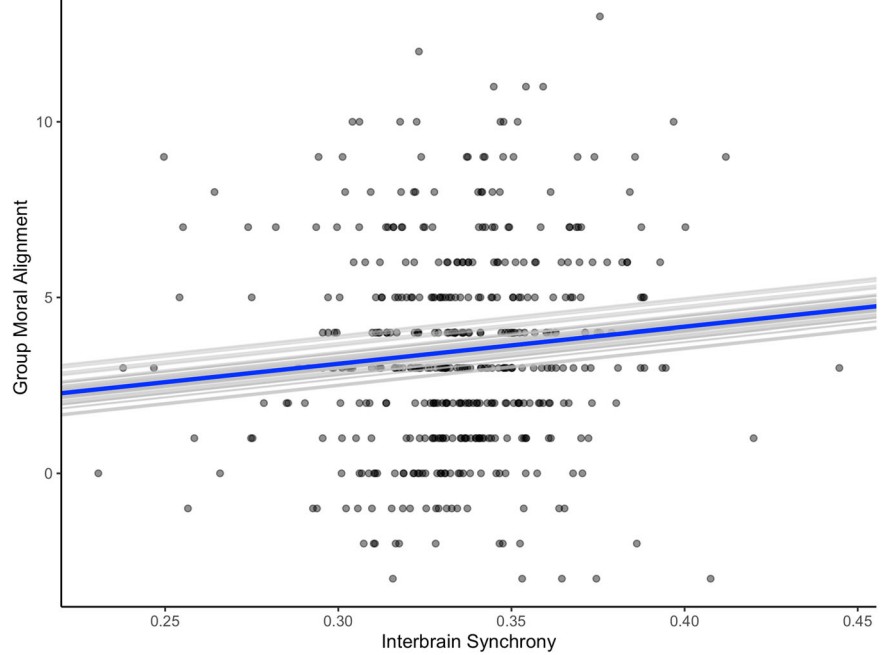

**Table 5 | Model comparison: the effect of left IFG interbrain synchrony on group moral alignment**

| Model | N parameters | BIC | Chi-square | DF | p value |
|---|---|---|---|---|---|
| Unconditional means model | 3 | 2255.7 | | | |
| Random intercept model | 4 | 2257.9 | 3.9333 | 1 | 0.047 |
| Random slope model | 5 | 2263.4 | 0.538 | 1 | 0.463 |

deliberation in small groups facing morally ambiguous dilemmas. Consistent with our predictions, we observed that group members adjusted their private moral judgments toward the group consensus following deliberation, particularly when interbrain synchrony in the left IFG was elevated during deliberations.

The behavioral results demonstrate that when confronted with morally ambiguous moral issues, group members whose views were initially not aligned tend to align their moral views after deliberations by systematically converging around the reached consensus, exhibiting higher alignment post-deliberation compared to pre-deliberation. Our findings are in line with recent research on moral conformity showing that moral beliefs are subject to external pressures to conform[7,8]. However, distinct from previous studies, the role of deliberation is central to our results. Rather than alignment arising passively from exposure to majority opinions, as seen in prior work[7,8], our study highlights how active exchange of arguments during deliberation drives convergence in moral judgments. This process is neither guaranteed nor uniform; deliberations with lower left IFG interbrain synchrony were less likely to yield alignment, suggesting that successful moral alignment hinges on effective communication and mutual understanding among group members. Our findings contribute to ongoing debates about the formation and persistence of moral norms in groups by providing evidence of the mechanisms through which alignment emerges during deliberation. The tendency among group members to align their views in the face of ambiguity may serve as the foundation of the development of moral norms among group members. Our results suggest that such moral norms may arise dynamically as a product of collective interactions, particularly when faced with ambiguity. This aligns with theories positing that moral norms are not merely static rules but develop through repeated negotiations[9,10]. Broadly, our research invites the field of moral psychology to expand its traditional scope, which primarily centers on individual moral reasoning and decision-making, by exploring moral decision-making in groups, as many real-life moral decisions, such as jury trials, ethics committees, war cabinets, and religious congregations demand collective deliberation and agreement[49].

In addition to demonstrating increased moral alignment following deliberation, the results of our neural analysis lend support to our hypothesis that the extent of moral alignment can be predicted by an increase in left IFG interbrain synchrony during deliberation. We first compared levels of interbrain synchrony during deliberation in real versus pseudo groups and showed that left IFG interbrain synchrony is higher in the case of groups engaging in actual conversation than in pseudo groups. This finding indicates that interbrain synchrony in the IFG during communication cannot be accounted for by confounding factors such as a common sensory environment or common cognitive processing of task-related features, but rather is specific to the interaction. Moreover, in line with our main hypothesis, we demonstrated that higher interbrain synchrony in the left IFG during deliberation predicted the higher alignment of privately held views toward the consensus. We also found that the effect is not influenced by group-level factors such as group gender composition or dominance structure, pointing to the relative robustness of our findings.

The prominence of left IFG interbrain synchrony in our results reinforces its role as a neural mechanism underpinning the alignment process and is in line with previous studies pointing to its role in facilitating coordination and communication. Previous research demonstrated increased interbrain synchrony in the left IFG during face-to-face dialog[39], coordinated singing[35], storytelling, and narrative conveying, with increased synchrony facilitating shared understanding[43,44]. Researchers have claimed that increased interbrain synchrony of the left IFG reflects synchrony of the same neural circuitry that underlies both production in speakers and comprehension in listeners, with increased synchrony corresponding to successful communication[23,35,39,43,44]. Together with previous studies, we highlight the role of interbrain synchrony in the left IFG between interacting humans both in effective communication and in developing a shared understanding that aligns divergent beliefs through the deliberation process[23,35,39,43,44]. By linking interbrain synchrony to deliberation outcomes, our study bridges social and neural phenomena, illustrating how alignment in moral conventions between group members is achieved through interactive processes.

The accumulating evidence for the role of left IFG interbrain synchrony in effective communication and coordination raises a question: What properties of the left IFG uniquely position it to support synchrony between brains and facilitate communication? While the left IFG was initially considered to be a core language production node[50], it has been recognized recently for its role in language comprehension through phonological, syntactic, and semantic processing[51], as well as the perception of biological motion, planning goal-directed actions, and processing narratives and music[52–55]. In an attempt to reconcile the involvement of the left IFG in such a wide range of processes, researchers have assigned it a domain-general function, namely dynamic sequence processing[53]. It has been suggested that a common denominator for all processes in which the left IFG is involved is the presence of stimuli composed of sequentially arranged elements, such as a series of coordinated movements constituting a purposeful motor program, or a sequence of lexical units comprising a meaningful sentence[53]. The left IFG is suggested to implement a binding mechanism responsible for unifying the sequential units into a meaningful whole[53]. It is possible that the IFG's role in sequence processing uniquely positions it to process sequentially structured information generated by others, such as their intentions, narratives, and arguments, with interbrain synchrony reflecting synchrony in the same neural circuitry underlying both production/encoding of such information in the sender and comprehension/decoding in the receiver[23,35,39,43,44].

A number of limitations of the current study are worth noting. First, the time limits of the deliberations may provoke a different process than what would otherwise be in play during natural deliberations in different environments. The time-constrained nature of deliberations in our study limits the external validity of our study. Second, our alignment measure reflects immediate changes in beliefs aligned with the consensus but does not capture potential long-term changes. Future research could explore whether moral alignment persists over time. Longitudinal studies would help determine if alignment in group contexts leads to lasting, generalizable shifts in individual beliefs or remains transient and context-dependent. Third, hyperscanning with fNIRS, though useful for studying multiple individuals simultaneously is sensitive to artifacts like variations in optical coupling and spatial resolution[34], which may introduce noise and limit the precision of interbrain synchrony measures. Fourth, the influence of cultural factors on moral judgments warrants further attention. Trolley-type dilemmas, while standard in moral psychology, reflect specific cultural assumptions and may resonate differently across cultures[56]. For instance, individualist cultures may prioritize personal responsibility, while collectivist cultures emphasize group harmony[57]. While our sample was relatively culturally heterogeneous consisting of both Arab- and Hebrew-speaking participants, future research should explore systematically how cultural differences impact moral alignment and its neural correlates through cross-cultural studies. Lastly, the mechanisms underlying interbrain synchrony are still poorly understood. Synchrony during communication can theoretically occur at multiple levels

https://doi.org/10.1038/s42003-025-07831-4                                   **Article**

of representation, ranging from low-level representations at the linguistic unit level (e.g., phonological, morphological, syntactic) to high-level representations at the semantic or conceptual unit levels (e.g., word, sentence, argument)[58]. Previous research reveals a gradient pattern in which low-level representation synchrony occurs in primary cortical regions, whereas high-level representation synchrony occurs at hierarchically higher cortical regions such as the left IFG[43]. The current study does not disentangle synchrony at different levels. Future studies may examine this issue using different paradigms. We also acknowledge the limitations of correlative brain-behavior studies in drawing causal inferences and addressing potential confounding factors that may influence both behavioral and neural measures. For example, while we found an association between increased left IFG interbrain synchrony and moral alignment, we cannot definitively establish whether the observed neural synchrony directly drives the behavioral alignment or whether both are influenced by a third variable, such as shared attention or mutual engagement during deliberation[31,37]. Our multi-level statistical modeling approach allowed us to assess the relationship between interbrain synchrony and moral alignment within each group thereby controlling for inter-group variability in baseline attention and engagement. Nonetheless, future studies employing causal methods, such as manipulating the degree of interbrain synchrony through experimental paradigms or neural interventions, are essential to help clarify the causal role of the left IFG in facilitating moral alignment during deliberation.

In conclusion, we demonstrate that interbrain synchrony in the left IFG during deliberations predicts their effectiveness in aligning moral views. This finding sheds light on how group alignment emerges from initially uncoordinated states and contributes to the broader investigation of factors that distinguish effective deliberation leading to belief and preference alignment, from ineffective deliberation. These insights hold significant implications for contexts where moral consensus is critical, such as jury deliberations, policy discussions, and ethical decision-making in organizations. Broadly, our results have important implications for research on collective decision-making, social influence, conflict management, and the future of deliberative democracy[1,59–61].

## Method
### Participants
A total of 200 healthy participants were randomly assigned to 50 groups of four participants per group. The desired sample size (groups n) was calculated using power analysis carried out in G*Power 3.1[62], with an assumed moderate effect size, $\alpha = 0.05$, and power of 0.80[63] to ensure the detection of significant effects in our main regression model. Three groups were later excluded from the analysis due to data acquisition problems, that were due to fNIRS system failures caused by Bluetooth connection failure or undiagnosed software shutdown. These failures resulted in an unsystematic loss of neuroimaging samples from participants in the excluded groups, preventing their inclusion in the final sample (47 groups; see Table 6 for participants' demographics).

General exclusion criteria included left-handedness—due to potential differences in brain lateralization that might introduce noise that affects neural measures—reading difficulties, and a history of neurological and/or psychiatric disorder. Participants within each group shared the same native language, either Hebrew or Arabic (N Arabic = 34 groups), ensuring the study was conducted in their mother tongue for optimal understanding of the material and effective expression of their opinions.

All ethical regulations relevant to human research participants were followed. The study was approved by the local Ethics Committee of the University of Haifa, and all participants signed an informed consent form prior to being admitted to the study.

### Procedure
Upon arriving at the lab, participants were randomly seated in front of personal computers facing each other (Fig. 1a), in a soundproof, well-lit lab with minimal visual distractions. They then were presented with standardized instructions and completed a moral judgment task, which comprises

**Table 6 | Demographic characteristics of participants (N = 188)**

| Characteristic | N | % |
|---|---|---|
| Gender | | |
| Female | 133 | 70.7446 |
| Male | 55 | 29.2553 |
| Age group (years) | | |
| 18–21 | 99 | 52.6595 |
| 22–25 | 63 | 33.5106 |
| 26–30 | 17 | 9.0425 |
| 31–39 | 9 | 4.7872 |
| Native language | | |
| Arabic | 136 | 72.3404 |
| Hebrew | 52 | 27.6595 |

Gender, age, and native language were self-reported by participants.

two stages (Fig. 1b). In Stage 1, participants were presented with a sequence of written descriptions of ten trolley-type moral dilemmas. Each dilemma describes a situation in which a protagonist is forced to decide between contrasting options: utilitarian (outcome-based) and deontological (principle-based). In each situation, a utilitarian decision was proposed to resolve the dilemma (e.g., kill one to save many). Participants were then asked to rate the moral appropriateness of the proposed decision on an eight-point Likert scale. The points on the scale were labeled and numbered, ranging from 1 (absolutely morally inappropriate) to 8 (absolutely morally appropriate). No point on the scale represented a neutral stance (see Supplementary Fig. 1 for a sample dilemma and rating scale). Each dilemma was presented in a separate screen and participants were able to progress from one screen to the next after rating each dilemma.

In Stage 2, participants were presented with the same sequence of ten moral dilemmas while their brain activity was scanned with fNIRS. Each dilemma appeared as part of a three-phase block including a fixation phase, a deliberation and collective rating phase, and a final private rating phase. In contrast to Stage 1 where participants could pass from one screen to another in a self-paced manner, in Stage 2, the four participants' screens were synchronized and progressed according to pre-determined timing. At the start of each dilemma's block, participants were presented with a fixation screen during which they were asked to remain silent and motionless and fixate on a cross shown at the center of their computer screens for 30 s. After the fixation phase, during the collective rating phase, the dilemma appeared on all four screens simultaneously for 140 s. In this time window, participants were asked to publicly share their initial views on the dilemma with each other and to engage in consensus-oriented deliberation to reach a collective rating (representing their consensus decision) during the allocated time. After the collective rating phase, and during the final private rating phase, the dilemma was again simultaneously shown on all four screens for 30 s during which participants assigned their final private rating to the dilemma. The task then progressed to the next dilemma's block and so forth. This procedure yielded three ratings for each participant per dilemma: an initial private rating in Stage 1 and a collective rating and a final private rating, both in Stage 2 (See Fig. 1b). The task was designed and administered via Qualtrics, which allowed for precise control over the presentation of moral dilemmas and standardized data collection across participants.

### Group moral alignment
The group moral alignment behavioral measure is computed as:

$$\text{Group Moral Alignment} = \sum_{i=1}^{N}(|R_{i,\text{initial}} - C| - |R_{i,\text{final}} - C|)$$

Where $N$ is the number of group members (in our case, four), $R_{i,\text{initial}}$ and $R_{i,\text{final}}$ are the initial and final private ratings of participant $i$, and $C$ represents

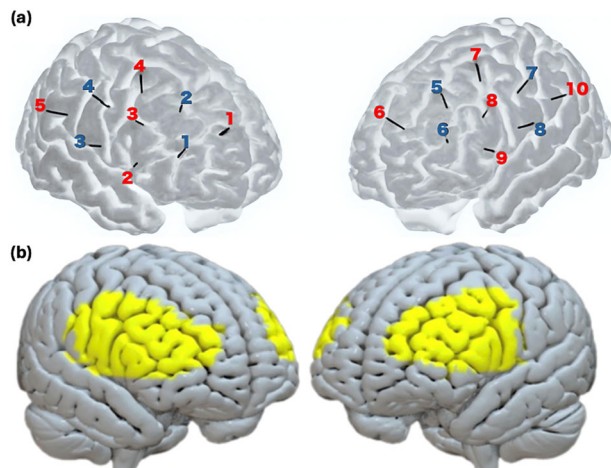

**Fig. 5 | Channels montage. a** Illustration of the Brite24 fNIRS system channels montage using AtlasViewer MATLAB application[87]. Red numbers are sources (five in each hemisphere), and blue numbers are detectors (four in each hemisphere); together they form 24 channels. Black lines represent optode projections from the scalp to the cortex. **b** Illustration of cortical surface captured by our montage (in yellow).

the consensus decision. By subtracting the difference between a participant's final rating and consensus from their initial rating and consensus, we calculate the degree to which the participant adjusted their private ratings towards the group consensus. The group moral alignment measure, therefore, captures the overall shift in group members' private moral views toward the group consensus.

### Neural data acquisition
We used the Brite24 fNIRS system (designed and described by Artinis Medical Systems, Elst, The Netherlands) to simultaneously measure changes in cortical oxygenated hemoglobin (O2Hb) and deoxygenated hemoglobin (HHb) concentrations. The Brite24 fNIRS system employs near-infrared light transmission at two wavelengths, 760 and 850 nm. Data were collected at a sampling frequency of 50 Hz. The system consists of a flexible probe unit (headset) with 18 optodes (10 sources and 8 detectors) forming 24 channels (12 in each hemisphere). The headset was positioned on the participants' heads according to the international 10–20 system, partially covering the bilateral prefrontal cortex (see Fig. 5 for more details on probe montage). Supplementary Table 8 provides the exact MNI coordinates for each optode in our montage, Supplementary Table 9 provides the exact MNI coordinates for each reference point in the 10–20 system.

### Implemented neural data preprocessing
Preprocessing was conducted using the HOMER3 MATLAB package[64] and included the following steps: (I) converting the light intensity data to optical density; (II) identifying motion artifacts and correcting them using a targeted principal component analysis approach[65] (III); applying a bandpass filter to the data (cutoff frequencies of 0–1 Hz) to help remove high-frequency components in the signal that are unrelated to brain activity[66]; (IV) converting the optical density data to O2Hb and HHb concentration values that capture cerebral blood flow (a proxy for neural activity) using the modified Beer–Lambert law[67]; (V) removing channels that exhibited a positive correlation (above 0.5) between O2Hb and HHb, based on the assumption that oxy (O2Hb) and deoxy (HHb) hemoglobin typically exhibit a strong negative correlation[68]. Scalp coupling index parameters were further calculated for each channel (Supplementary Data 1).

　　　　Channels were then clustered into six ROIs by averaging their preprocessed time series into one time series per ROI. The clustering into ROIs was based on the estimated Broadmann area above which they were placed. The Broadmann area corresponding to each channel was estimated by

connecting the digitized Montreal Neurological Institute (MNI) coordinates of each channel to the location of that channel according to the 10–20 system. The six ROIs include the right and left IFG, right and left dorsolateral prefrontal cortex (DLPFC), and right and left premotor cortex (PMC). Supplementary Table 10 specifies which source and detector compose each channel and which channels compose each ROI.

### Wavelet transform coherence
The wavelet transform coherence (WTC) approach was used to assess interbrain synchrony based on the coherence between two time series at any relevant time and frequency range[69]. Coherence values range from 0 (no coherence) to 1 (complete coherence). We applied WTC to the two neural signal time series generated from each corresponding pair of ROIs across brains (e.g., Left IFG in Participant One and Left IFG in Participant Two) and focused on the frequencies ranging from 0.16 Hz (period 6 s), corresponding to the typical 6-s lag in the hemodynamic response[70], to 0.015 Hz (period 66 s). This frequency range excludes artifacts related to breathing (~0.2–0.3 Hz.) and heart rate (~1–2 Hz) and includes the frequency range suggested to optimize the wavelet coherence analysis as a measure of neural synchrony in fNIRS hyperscanning studies[71]. WTC was implemented using the wavelet coherence package in MATLAB[69].

### Group-level interbrain synchrony calculation
For each dyad of participants in a given group, we calculated a dyadic interbrain synchrony measure for each of the six homologous ROI pairs across brains, by aggregating coherence over the entire frequency range and time window of each deliberation phase. For each deliberation out of the ten, we then averaged the dyadic interbrain synchrony values to extract a group-level interbrain synchrony value. This process yields one group-level interbrain synchrony value per deliberation phase (out of ten) and per ROI pairing (out of six).

### Pseudo groups
To rule out the possibility that interbrain synchrony levels are fully accounted for by the fact that group members perform the same task and at the same time and not to the interaction itself, we compared real groups to pseudogroups created by randomly grouping participants who took part in the experiment as members of different groups. We generated 56 pseudo groups, ensuring that members of those pseudo groups, like real groups, shared a native language (the shuffling process to create random pairs occurs between groups that share the same native language), and ensuring that no two pseudo groups share a common pair of participants.

### Statistics and reproducibility
We used HLM to model the relationship between the independent variable group-level interbrain synchrony and the outcome variable group moral alignment, allowing either the intercept alone or the intercept and slope to vary by group and by item (item in our case refer to a specific dilemma out of the ten discussed). Modeling the relationship between the two variables using HLM allowed us to control for statistical problems that might arise when dealing with nested data structures[72,73]. Model fitting was conducted using the maximum likelihood estimation method utilizing the lme4 package in the R software environment[74,75].

　　　　Note that we focused our statistical analyses on the HHb signal. Nevertheless, we also applied the same analyses to the O2Hb signal and reported the results. Our decision to focus on the HHb signal is based on previous research showing that compared to O2Hb, the HHb signal is characterized by lower signal-to-noise and is less sensitive to physiological factors such as blood pressure, respiration and blood flow that can confound the hemodynamic response estimation[76–80]. Moreover, the HHb signal is more closely related to the HHb acquired by fMRI[81] and is more commonly used in fNIRS work that investigates interbrain synchrony during human face-to-face conversation[82]. fNIRS studies confirm that changes in arterial $CO_2$ while speaking alter the O2Hb signal to a greater extent than the HHb signal[83–85].

Across all implemented HLMs, Kolmogorov–Smirnov tests were conducted on model residuals to test the assumptions of normality. Additionally, Variance Inflation Factors (VIFs) were calculated to ensure that multicollinearity was not a concern in a model with more than one predictor (when VIF = 1).

## Reporting summary

Further information on research design is available in the Nature Portfolio Reporting Summary linked to this article.

## Data availability

All primary datasets and referenced datasets are publicly available at the Mendeley Data repository (https://doi.org/10.17632/ks8c66myft.1)[86].

## Code availability

All costume codes analysis scripts are publicly available at [https://doi.org/10.17632/ks8c66myft.1][86].

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

## Acknowledgements
This publication was supported by the European Research Council (ERC) under the European Union's Horizon 2020 Research and Innovation Programme (grant agreement no. INTERPLASTIC: 101020091; DLV-101020091).

## Author contributions
The authors confirm their contribution to the paper as follows: conceptualization: Sobeh and Shamay-Tsoory; methodology: Sobeh and Shamay-Tsoory; investigation: Sobeh; formal analysis: Sobeh and Shamay-Tsoory; writing—original draft preparation: Sobeh and Shamay-Tsoory. All authors reviewed the results and approved the final version of the manuscript.

## Competing interests
The authors declare no competing interests.
