## [Transparent Peer Review file · Communications Biology]

The Emergence of Moral Alignment Within Human Groups is Facilitated by Interbrain Synchrony

Corresponding Author: Mr Aial Sobeh

Version 0:

Reviewer comments:

Reviewer #1

(Remarks to the Author)

Thank you for the opportunity to review this well-conceptualised and equally well-written manuscript.

The major claims of the manuscript and the findings thereof, is that moral alignment within human groups is largely facilitated by interbrain synchrony – which, in this case, is synchronised in the left inferior frontal gyrus (IFG). Expanding on previous research on interbrain synchrony, as localised to the IFG, the manuscript introduces an element of novelty, by investigating information exchange, between deliberating partners and moral belief alignment, using behavioural and neuroimaging measures (fNIRS). The study methods are clearly detailed, and the statistical analysis well detailed and executed, to allow for study replicability. To this end, the Mendeley Data provided by the authors, in conjunction with the methods section, is exhaustive to allow for replicability.

Notwithstanding, the article could be strengthened by addressing the following:

Page 3 - Paragraph 2, lines 7-10 – this is an essential piece of text that could be revised to enhance readability.

Methods section

It is unclear why the authors excluded left-handed participants; this could be explained better in the Discussion section (page 6).

Although a moderate effect size is assumed, I would suggest the authors provide the exact effect size measure (d , η^2 , etc) used in the power analysis.

The authors state that three groups were excluded from the final analysis due to data acquisition problems – again, for replicability, I would suggest the authors state what these problems were (e.g., fNIRS system failure, physiological noise, etc.).

I would recommend that the authors include a table indicating the demographic data of the research participants.

The procedure section is well written and supplemented by the Supplementary Materials section.

Behavioural measures

The authors explain how they calculated group moral judgment – as helpful as this is in the word form – I would suggest they perhaps provide a simple formulaic description of the calculation.

fNIRS data pre-processing:

In addition to the provided pre-processing steps, I would suggest the authors provide data on the scalp coupling index (SCI) parameters. Where participants excluded due to SCI violations?

Page 11, paragraph 2, line 4 – I think ‘average’ should read ‘averaged’.

I commend the authors for the novelty of comparing real groups to pseudo groups and investigating group-level factors (i.e.,

gender composition and group dominance structure) and how these could impact moral alignment.

Conclusion:

I think the conclusion section could be enhanced by briefly discussion of (a) the limitations of behavioural measures when sometime juxtaposed with neuronal measures (e.g., fNIRS), (b) the data acquisition problems encountered, and if these are related to fNIRS instrumentation, and (c) the influence / role of culture in moral judgements, especially in individualistic cultures.

Overall, I think this is a well-researched and well-written manuscript, and I recommend it for publication with the above issues addressed.

Kind regards,

Reviewer #2

(Remarks to the Author)

The manuscript explores a compelling and innovative topic: the role of interbrain synchrony, specifically in the left inferior frontal gyrus (IFG), in facilitating moral alignment during group deliberations. The combination of behavioral and neuroimaging methods (fNIRS) is a strength point of the manuscript, providing both theoretical insights and empirical contributions to the fields of moral psychology and social neuroscience. However, several critical issues should be addressed to improve clarity, rigor, and the interpretative nature of the study.

Major points:

-Introduction

The introduction presents the concept of interbrain synchrony well but does not sufficiently distinguish how this study extends prior work. It would benefit from a clearer explanation of how moral alignment and interbrain synchrony interact uniquely compared to other forms of social alignment studied in the literature.

The discussion of "moral norms" lacks nuance. The manuscript could explore how the findings contribute to ongoing debates about the formation and persistence of moral norms in groups.

-Method

Participants: The exclusion of three groups due to "data acquisition problems" is mentioned but not elaborated. A clearer explanation of how these exclusions might affect the representativeness or generalizability of the findings is needed.

Tasks and Measures: The use of the trolley dilemma is a classic choice but introduces potential biases due to its cultural and contextual variability. How do the authors ensure that these dilemmas are equally relevant to participants with different cultural backgrounds?

fNIRS Measures: While the preprocessing steps are detailed, it is unclear whether potential confounding variables (e.g., physiological differences between participants, noise in the lab environment) were fully controlled. How were these factors controlled?

-Statistical Analyses

The hierarchical linear modeling approach is appropriate, but key details are missing:

Were multicollinearity or other statistical assumptions tested before applying the models?

The justification for focusing primarily on the HHb signal should be more rigorously tied to prior empirical evidence, given the absence of significant findings in O2Hb-based analyses.

-Discussion

The interpretation of left IFG synchrony as a mechanism for moral alignment is plausible but appears overly deterministic. Could alternative explanations, such as cognitive effort or task-related engagement, account for the observed patterns?

The manuscript does not sufficiently address whether moral alignment is transient or persists over time. A discussion of potential longitudinal implications would be valuable.

-Limitations

The limitations section is relatively brief. For instance, the artificial nature of the experimental setting (e.g., time-constrained deliberations) is a notable limitation that could affect external validity. The authors should also consider whether dominance dynamics or conversational styles influenced the results.

Figures and Tables:

Figures 3 and 4 are informative but lack sufficient labeling and contextual explanation. It would be helpful to provide more intuitive interpretations of the axes and data points for readers unfamiliar with fNIRS coherence metrics.

Specific Comments:

Figure 1: To allow reproducibility, please provide the exact correspondence between sources and detectors location and the 10-20 or 10-10 or 10-5 international system. Also please specify from which injector and detector each channel is composed.

Page 7: Clarify the rationale for selecting groups of four participants. Was this choice based on theoretical grounds or practical considerations?

Page 14: The explanation of the moral alignment measure is somewhat convoluted. Simplify and ensure it is comprehensible to readers without advanced statistical expertise.

Page 27: The discussion refers to "high conceptual synchrony," but this term is not explicitly defined or grounded in prior literature.

The manuscript addresses an important and underexplored area of social neuroscience. However, revisions are necessary to refine theoretical framing, strengthen methodological rigor, and ensure clarity in the presentation and interpretation of findings. Addressing these issues will significantly enhance the manuscript's contribution to the field.

Version 1:

Reviewer comments:

Reviewer #2

(Remarks to the Author)

The authors have successfully fulfilled my requests for revisions.

Response to Reviewers

Reviewer #1

Comment 1: *Page 3 - Paragraph 2, lines 7-10 – this is an essential piece of text that could be revised to enhance readability.*

Response: We have revised the specified text to elaborate on the importance of a multi-brain approach to the study of social alignment, and enhance its readability. The updated version can be found on page 3-4, paragraph 2, lines 7-16 :

“... examining the interbrain relationships during communication can illuminate when deliberation is likely to align beliefs among group members (1). Yet, prior neuroimaging research on social alignment has typically examined brain activity under conditions that restricted communication. For instance, participants were often tasked with expressing their preferences (e.g., rating the attractiveness of an item or agreeing with a statement) before being exposed to an experimentally manipulated group influence, such as a majority opinion differing from their own (26–29). These paradigms allowed researchers to investigate the neural activity associated with changes in personal attitudes toward the group’s preference (a measure of alignment) following exposure to social information. However, by restricting verbal expression and direct social interaction, these studies offer a limited view of the dynamic processes through which alignment emerges during social deliberation. A full account of the process through which group members align their thoughts requires capturing how multiple brains communicate to influence and infer each other’s mental states (2,23,30).”

Comment 2: *It is unclear why the authors excluded left-handed participants; this could be explained better in the Discussion section (page 6).*

Response: We thank the reviewer for bringing our attention to the need of clarifying our choice regarding the exclusion of left-handed participants. We now explain that left-handed participants were excluded due to potential differences in brain lateralization that might introduce noise and thus affecting interbrain synchrony measures. This clarification is now discussed in the Method section (section 2.1.):

“ General exclusion criteria included left-handedness — due to potential differences in brain lateralization that might introduce noise that affect neural measures “

Comment 3: *Although a moderate effect size is assumed, I would suggest the authors provide the exact effect size measure (d, =, etc) used in the power analysis.*

Response: We have updated parts of results section to specify the exact effect size. This information is now included to the analyses reported in section 3.1 and section 3.3.

Comment 4: *The authors state that three groups were excluded from the final analysis due to data acquisition problems – again, for replicability, I would suggest the authors state what these problems were (e.g., fNIRS system failure, physiological noise, etc.).*

Response: As suggested, we now explain in the method section the reasons for excluding these groups. We specify that the data acquisition problems were due to fNIRS system failures caused by various factors. This detail is provided in the Method section (section 2.1.):

“ Three groups were later excluded from the analysis due to data acquisition problems, that were due to fNIRS system failures caused by Bluetooth connection failure or undiagnosed software shutdown. These failures resulted in an unsystematic loss of neuroimaging samples from participants in the excluded groups, preventing their inclusion in the final sample “

Comment 5: *I would recommend that the authors include a table indicating the demographic data of the research participants.*

Response: As suggested, a table summarizing participant demographic data has been added to the manuscript (Table 1). All following tables were renumbered accordingly.

Table 1.

Demographic Characteristics of Participants (N= 188)

Characteristic	N	%
Gender		
Female	133	70.7446
Male	55	29.2553
Age Group (years)		
18–21	99	52.6595
22–25	63	33.5106
26-30	17	9.0425
31-39	9	4.7872
Native Language		
Arabic	136	72.3404
Hebrew	52	27.6595

Note: Gender, Age, and Native language were self-reported by participants.

Comment 6: *The authors explain how they calculated group moral judgment – as helpful as this is in the word form – I would suggest they perhaps provide a simple formulaic description of the calculation.*

Response: We thank the reviewer for their comment. As suggested, we have now updated our description of the calculation and included a formulaic expression of the group moral alignment calculation in the Methods section (subsection 2.3.1):

“The group moral alignment measure is computed as:

$$\text{Group Moral Alignment} = \frac{1}{N} \sum_{i=1}^N (|R_{i,initial} - C| - |R_{i,final} - C|)$$

Where N is the number of group members (in our case, four), $R_{i,initial}$ and $R_{i,final}$ are the initial and final private ratings of participant i , and C represents the consensus decision. By subtracting the difference between a participant's final rating and consensus from their initial rating and consensus, we calculate the degree to which the participant adjusted their private ratings towards the group consensus. The group moral alignment measure, therefore, captures the average shift in participants' private moral views toward the group consensus. “

Comment 7: *In addition to the provided pre-processing steps, I would suggest the authors provide data on the scalp coupling index (SCI) parameters. Were participants excluded due to SCI violations?*

Response: We thank the reviewer for their insightful suggestion regarding the scalp coupling index (SCI) parameters. As outlined in our method section, channels were excluded during pre-processing based on the correlation between oxygenated and deoxygenated blood concentration, not SCI violations.

In response to your suggestion, we have now calculated SCI values and included them in a Supplementary Data Excel file (Supplementary Data 1), which will be submitted with the revised manuscript. For clarity, our original method led to the rejection of 365 channels out of 4512 which is similar to the SCI method (where 426 channels would have been excluded).

Comment 8: *Page 11, paragraph 2, line 4 – I think ‘average’ should read ‘averaged’.*

Response: We apologize for this spelling error. The text has been corrected to ‘averaged’ on page 11.

Comment 9: *The conclusion section could be enhanced by briefly discussing (a) the limitations of behavioral measures when juxtaposed with neuronal measures, (b) the data acquisition problems encountered, and (c) the influence/role of culture in moral judgments, especially in individualistic cultures.*

Response: We appreciate the reviewer’s suggestion to expand the limitations section. We now elaborate on the data acquisition challenges, the influence of cultural factors on moral judgments, and the constraints of the study (and correlative brain-behavior research in general) in drawing causal inferences and addressing potential confounding factors.

“ Third, hyperscanning with fNIRS, though useful for studying multiple individuals simultaneously, is sensitive to artifacts like variations in optical coupling and spatial resolution (34), which may introduce noise and limit the precision of interbrain synchrony measures. Fourth, the influence of cultural factors on moral judgments warrants further attention. Trolley-type dilemmas, while standard in moral psychology, reflect specific cultural assumptions and may resonate differently across cultures (76). For instance, individualist cultures may prioritize personal responsibility, while collectivist cultures emphasize group harmony (77). While our sample was relatively culturally heterogeneous consisting of both Arab and Hebrew speaking participants, future research should explore

systematically how cultural differences impact moral alignment and its neural correlates through cross-cultural studies”.

“We also acknowledge the limitations of correlative brain-behavior studies in drawing causal inferences and addressing potential confounding factors that may influence both behavioral and neural measures. For example, while we found an association between increased left IFG interbrain synchrony and moral alignment, we cannot definitively establish whether the observed neural synchrony directly drives the behavioral alignment or whether both are influenced by a third variable, such as shared attention or mutual engagement during deliberation (31,37). Future studies employing causal methods, such as manipulating the degree of interbrain synchrony through experimental paradigms or neural interventions, could help clarify the causal role of the left IFG in facilitating moral alignment during deliberation.”

Reviewer #2

Comment 1: *The introduction presents the concept of interbrain synchrony well but does not sufficiently distinguish how this study extends prior work. It would benefit from a clearer explanation of how moral alignment and interbrain synchrony interact uniquely compared to other forms of social alignment studied in the literature.*

Response: We appreciate the reviewer’s insightful comment. As suggested, we have expanded our discussion on the novel theoretical rationale behind this study. In the Introduction section, we clarify our unique approach to investigating the neural processes underlying social alignment—specifically, through a multi-brain perspective using hyperscanning to examine interbrain synchrony (This can be found on pages 3-4, paragraph 2, lines 7-16, and page 5, paragraph 2, lines 1-2.) (see the first paragraph below). In particular, we emphasize that while prior research on social alignment has primarily focused on individual brain activity linked to aligning with majority opinions, our study is the first to explore how social alignment at the collective level emerges through group deliberation—starting from initial states of misalignment and without relying on pre-established majority views. Furthermore, we elaborate on our decision to examine social

alignment in moral judgments, as it allows us to investigate the interactive emergence of moral alignment through deliberation. This contrasts with previous research on moral change, which has largely focused on individuals updating their moral beliefs following exposure to social information. (This can be found on page 2, paragraph 2, lines 1-5) (see the second paragraph below):

“... examining the interbrain relationships during communication can illuminate when deliberation is likely to align beliefs among group members (1). Yet, prior neuroimaging research on social alignment has typically examined brain activity under conditions that restricted communication. For instance, participants were often tasked with expressing their preferences (e.g., rating the attractiveness of an item or agreeing with a statement) before being exposed to an experimentally manipulated group influence, such as a majority opinion differing from their own (26–29). These paradigms allowed researchers to investigate the neural activity associated with changes in personal attitudes toward the group’s preference (a measure of alignment) following exposure to social information. However, by restricting verbal expression and direct social interaction, these studies offer a limited view of the dynamic processes through which alignment emerges during social deliberation. A full account of the process through which group members align their thoughts requires capturing how multiple brains communicate to influence and infer each other’s mental states (2,23,30) ... The current study investigated the interbrain relationships during communication that can predict the emergence of moral alignment”

“Recent studies in the field of moral psychology show that the human tendency to align extends to moral decision-making, with individuals adjusting their moral beliefs to align with what they perceive as the norm (7,8) While previous research on moral change provides evidence for the flexibility of moral views and the tendency of individuals to align with perceived majority opinions (7,8), these studies largely overlook the interactive processes through which humans arrive at moral agreements. Specifically, little is known about how social deliberation—a collaborative and dynamic form of interaction—drives moral agreement within groups faced with morally ambiguous dilemmas. “

Comment 2: *The discussion of “moral norms” lacks nuance. The manuscript could explore how the findings contribute to ongoing debates about the formation and persistence of moral norms in groups.*

Response: We have expanded the discussion of moral norms, connecting our findings to broader debates on their formation and persistence within groups. This revision is now reported in the Discussion section, page 28, lines 5-11:

“Our findings contribute to ongoing debates about the formation and persistence of moral norms in groups by providing evidence of the mechanisms through which alignment emerges during deliberation. The observed tendency for group members to converge on a consensus suggests that moral norms may arise dynamically as a product of collective interactions, particularly when faced with ambiguity. This aligns with theories positing that moral norms are not merely static rules but develop through repeated negotiations (9,10). Our research invites the field of moral psychology to expand its traditional scope, which primarily centers on individual moral reasoning and decision-making, by exploring moral decision-making in groups, as many real-life moral decisions, such as jury trials, ethics committees, war cabinets, and religious congregations demand collective deliberation and agreement (82).”

Comment 3: *The exclusion of three groups due to “data acquisition problems” is mentioned but not elaborated.*

Response: We thank the reviewer for this comment. We have now elaborated on the reasons for these exclusions in the Method section (section 2.1.):

“ Three groups were later excluded from the analysis due to data acquisition problems, that were due to fNIRS system failures caused by bluetooth connection failure or undiagnosed software shutdown. These failures resulted in an unsystematic loss of

neuroimaging samples from participants in the excluded groups, preventing their inclusion in the final sample “

Comment 4: *The use of the trolley dilemma is a classic choice but introduces potential biases due to its cultural and contextual variability. How do the authors ensure that these dilemmas are equally relevant to participants with different cultural backgrounds?*

Response: As suggested we now address the issue of cultural differences in the discussion section. As we explain, we ensured that the dilemmas were equally relevant to participants from both Jewish and Arab (specifically Muslim) cultural backgrounds by conducting a pilot study with both groups to confirm the consistent interpretation of the scenarios. Additionally, we selected dilemmas that address universally applicable ethical principles, such as harm, public safety, and fairness, while avoiding culturally specific elements. Nonetheless we acknowledge the limitations of this paradigm in the Limitations section:

“Fourth, the influence of cultural factors on moral judgments warrants further attention. Trolley-type dilemmas, while standard in moral psychology, reflect specific cultural assumptions and may resonate differently across cultures (76). For instance, individualist cultures may prioritize personal responsibility, while collectivist cultures emphasize group harmony (77). While our sample was relatively culturally heterogeneous consisting of both Arab and Hebrew speaking participants, future research should explore systematically how cultural differences impact moral alignment and its neural correlates through cross-cultural studies”

Comment 5: *It is unclear whether potential confounding variables (e.g., physiological differences between participants, noise in the lab environment) were fully controlled. How were these factors controlled?*

Response: To address this comment we explain in the method section that when applying the The Wavelet Transform Coherence (WTC) approach — to assess interbrain synchrony based on the coherence between two time-series at any relevant time and frequency range — we focused on a frequency range the excludes artifacts related to breathing and heart rate, thereby partly controlling for physiological noise (This information is included in the Methods subsection 2.4.3.) Furthermore, we now explain in the method section that to control for environmental noise, we conducted the study in a soundproof, well-lit lab with minimal visual distractions (This information is included in the Methods subsection 2.2.). We also now explain that to address procedural noise, we standardized instructions across participants, maintained consistent testing conditions (e.g., time of day was always between 12:00 and 18:00 PM), and used automated stimulus presentation software (Qualitrics) to eliminate variability in stimulus delivery (Information regarding the use of standardized instructions and automated stimulus presentation software are now included in the Methods subsection 2.2.).

Comment 6: *Were multicollinearity or other statistical assumptions tested before applying the models?*

Response: As suggested we now explain in the method section that we addressed multicollinearity and other statistical assumptions by testing prior to applying the hierarchical linear models (HLMs). Variance inflation factors (VIFs) were calculated to ensure that multicollinearity was not a concern (when $VIF = 1$) in models including more than one predictor. Additionally, residual diagnostics were performed to confirm the assumption of normality. Specifically, we employed Kolmogorov-Smirnov (K-S) test. Across all implemented HLM and linear regression analyses, the K-S test returned a non-significant p-value, supporting the assumption of normality. Details of these checks have been added to the Statistical Analyses section (section 2.5).

Comment 7: *The justification for focusing primarily on the HHb signal should be more rigorously tied to prior empirical evidence, given the absence of significant findings in O2Hb-based analyses.*

Response: We explain that we focus on the HHb signal based on previous research showing that compared to O2Hb, the HHb signal is characterized by lower signal-to-noise and is less sensitive to physiological factors such as blood pressure, respiration and blood flow that can confound the hemodynamic response estimation (1-5). Moreover, the HHb signal is more closely related to the HHb acquired by fMRI (6) and is more commonly used in fNIRS work that investigates interbrain coupling during human face-to-face conversation (7). This information is included in the Statistical Analyses section (section 2.5).

References:

- 1- Strangman, G., Culver, J. P., Thompson, J. H., & Boas, D. A. (2002). A quantitative comparison of simultaneous BOLD fMRI and NIRS recordings during functional brain activation. *Neuroimage*, 17(2), 719-731.
- 2- Kirilina, E., Jelzow, A., Heine, A., Niessing, M., Wabnitz, H., Brühl, R., ... & Tachtsidis, I. (2012). The physiological origin of task-evoked systemic artefacts in functional near infrared spectroscopy. *Neuroimage*, 61(1), 70-81.
- 3- Tachtsidis, I., & Scholkmann, F. (2016). False positives and false negatives in functional near-infrared spectroscopy: issues, challenges, and the way forward. *Neurophotonics*, 3(3), 031405-031405.
- 4- Zhang, X., Noah, J. A., & Hirsch, J. (2016). Separation of the global and local components in functional near-infrared spectroscopy signals using principal component spatial filtering. *Neurophotonics*, 3(1), 015004-015004.
- 5- Boas, D. A., Dale, A. M., & Franceschini, M. A. (2004). Diffuse optical imaging of brain activation: approaches to optimizing image sensitivity, resolution, and accuracy. *Neuroimage*, 23, S275-S288.
- 6- Ogawa, S., Lee, T. M., Kay, A. R., & Tank, D. W. (1990). Brain magnetic resonance imaging with contrast dependent on blood oxygenation. *proceedings of the National Academy of Sciences*, 87(24), 9868-9872.
- 7- Hirsch, J., Tiede, M., Zhang, X., Noah, J. A., Salama-Manteau, A., & Biriotti, M. (2021). Interpersonal agreement and disagreement during face-to-face dialogue: an fNIRS investigation. *Frontiers in human neuroscience*, 14, 606397.

Comment 8: *The interpretation of left IFG synchrony as a mechanism for moral alignment is plausible but appears overly deterministic. Could alternative explanations, such as cognitive effort or task-related engagement, account for the observed patterns?*

Response: We thank the reviewer for bringing the need for highlighting alternative explanations to our attention. To account for potential alternative explanations, such as baseline differences in attention or task-related engagement across groups, we employed multi-level statistical modeling with group ID as a random factor. This approach allowed us to assess the relationship between interbrain synchrony and moral alignment within each group while presumably controlling for inter-group variability in baseline attention and engagement. Nonetheless, we acknowledge the potential for alternative explanations and have included a discussion of how shared attention or mutual engagement could influence the observed correlation between interbrain synchrony and moral alignment. We indicate that we invite future studies to employ causal methods to test the suggested role of IFG synchrony as a mechanism facilitating the emergence of moral alignment through deliberation. This issue is now addressed in the Limitation paragraph of the Discussion section (see paragraph below).

“ Lastly, we acknowledge the limitations of correlative brain-behavior studies in drawing causal inferences and addressing potential confounding factors that may influence both behavioral and neural measures. For example, while we found an association between increased left IFG interbrain synchrony and moral alignment, we cannot definitively establish whether the observed neural synchrony directly drives the behavioral alignment or whether both are influenced by a third variable, such as shared attention or mutual engagement during deliberation (31,37). Our multi-level statistical modeling approach allowed us to assess the relationship between interbrain synchrony and moral alignment within each group thereby controlling for inter-group variability in baseline attention and engagement. Nonetheless, future studies employing causal methods, such as manipulating the degree of interbrain synchrony through experimental paradigms or neural interventions, are essential to help clarify the causal role of the left IFG in facilitating moral alignment during deliberation.

Comment 9: *The manuscript does not sufficiently address whether moral alignment is transient or persists over time. A discussion of potential longitudinal implications would be valuable.*

Response: We appreciate the Reviewer's important comment regarding the durability and long-term stability of the reported effects. We now explain that the effect of moral alignment were measured immediately following the deliberations which limits our ability to assess long-term alignment. We now address this issue in the Limitation section, and discuss the potential longitudinal implications of moral alignment in the revised manuscript. Specifically, we have clarified that our measure captures immediate changes in beliefs in line with the group consensus but does not assess whether these changes persist beyond the experimental context. We have also highlighted the need for future longitudinal studies that would examine whether moral alignment leads to lasting shifts in individual beliefs or if such changes are transient. This revision is in the Limitation paragraph of the Discussion section:

“ Second, our alignment measure reflects immediate changes in beliefs aligned with the consensus but does not capture potential long-term changes. Future research could explore whether moral alignment persists over time. Longitudinal studies would help determine if alignment in group contexts leads to lasting, generalizable shifts in individual beliefs or remains transient and context dependent.”

Comment 10: *The limitations section is relatively brief. For instance, the artificial nature of the experimental setting (e.g., time-constrained deliberations) is a notable limitation that could affect external validity. The authors should also consider whether dominance dynamics or conversational styles influenced the results.*

Response: We thank the reviewer for this comment. We have expanded the Limitations section to: (1) elaborate on the artificial nature of deliberations when time-constrained and its potential effects on external validity. (2) The limitation of the study in that it does not address potential longitudinal implications of moral alignment. (3) The influence of cultural assumptions on moral judgments

(4) The limitation of the correlative research approach in drawing causal inferences and addressing potential confounding factors. These revisions are now reported in the Limitation paragraph of the Discussion section. Furthermore, we address the effect of dominance dynamics on the relationship between interbrain synchrony and moral alignment in the Results section (section 3.3.):

“... we examined whether the slopes were dependent on group dominance structure by running a Spearman’s correlation test. Group dominance structure (see supplementary method section 1.1) captures the dominance equality between group members by measuring the variance of dominance scores between group members (i.e., a “completely egalitarian” group would have zero variance). The results of the Spearman’s correlation test indicated no significant correlation between the two variables, ($r(45) = .246, p = .094$).”

Comment 11: *Figures 3 and 4 are informative but lack sufficient labeling and contextual explanation. It would be helpful to provide more intuitive interpretations of the axes and data points for readers unfamiliar with fNIRS coherence metrics.*

Response: As suggested we now improve the Figure captions throughout the manuscript. Specifically, Figures 3 and 4 have been re-written with improved labeling and additional explanatory text in the figure captions to aid interpretation for readers unfamiliar with interbrain synchrony metrics.

Comment 12: *To allow reproducibility, please provide the exact correspondence between sources and detectors location and the 10-20 or 10-10 or 10-5 international system. Also please specify from which injector and detector each channel is composed.*

Response: We now include three supplementary tables (Table S1, and Table S2, and Table S3). Due to the fNIRS signal- optimization requirement of maintaining three centimeters between each source and detector, we were not able to ensure exact correspondence between our montage and

the 10-20 system. However, we provide exact MNI coordinates for optode placement and for the 10-20 reference points. Supplementary Table 1 provides the exact MNI coordinates for each optode in our montage, Supplementary Table 2 provides the exact MNI coordinates for each reference point in the 10-20 system, and Supplementary Table 3 specifies which source and detector compose each channel. The tables are now referenced in the main manuscript and included in the supplementary materials document.

Comment 13: *Clarify the rationale for selecting groups of four participants. Was this choice based on theoretical grounds or practical considerations?*

Response: To address the issue of group size, we explain that previous findings suggest that group size plays a critical role in shaping group dynamics and the emergence of synchrony-related phenomena (Tarr et al., 2016). Notably, synchrony in larger groups enables the formation of complex communication networks and subgroup dynamics that are not typically observed in dyadic interactions. Building upon these findings, the present study specifically examines groups of four participants to explore these dynamics further. This group size is sufficiently large to capture variations in moral reasoning and group deliberation, while still allowing for active engagement from each participant. It also aligns with the technical requirements of hyperscanning paradigms using functional near-infrared spectroscopy (fNIRS), which necessitates simultaneous monitoring of multiple participants while maintaining the quality of brain activity data.

We appreciate the reviewer's suggestion to include a clarification regarding the choice of group size. We now provide a clarification in the Introduction section.

“Previous research highlights the critical role of group size in shaping dynamics and synchrony-related phenomena (86). In groups, synchrony fosters complex communication networks and subgroup dynamics that are absent in dyadic interactions. Additionally, studies on small-group behavior in mammals show that groups of four exhibit movement patterns that mirror the underlying social interaction maps and dynamics of the group (87). Building on these findings, this study focuses on groups of four to explore these dynamics further.”

Comment 14: *The explanation of the moral alignment measure is somewhat convoluted. Simplify and ensure it is comprehensible to readers without advanced statistical expertise.*

Response: We thank the reviewer for their suggestion. We have updated our description and included a formulaic expression of the group moral alignment calculation in the Methods section (subsection 2.3.1):

“The group moral alignment measure is computed as:

$$\text{Group Moral Alignment} = \frac{1}{N} \sum_{i=1}^N (|R_{i,initial} - C| - |R_{i,final} - C|)$$

Where N is the number of group members (in our case, four), R_i initial and R_i final are the initial and final private ratings of participant i , and C represents the consensus decision. By subtracting the difference between a participant's final rating and consensus, from their initial rating and consensus, we calculate the degree to which the participant adjusted their private ratings towards the group consensus. The group moral alignment measure, therefore, captures the average shift in participants' private moral views toward the group consensus. “

Comment 15: *The discussion refers to “high conceptual synchrony,” but this term is not explicitly defined or grounded in prior literature.*

Response: Conceptual synchrony is a term previously used (reference 1) to refer to a condition in which individuals' mental representations have become aligned. We recognize the vagueness of the term and therefore drop its use altogether. Instead, we reiterated the suggestion that interbrain synchrony in the IFG reflects coupling in the neural circuitry underlying both the production and observation systems of interacting partners, and we highlight the fact that we did not disentangle

synchrony at different levels of representation as a clear limitation of our study. This revision is in the Limitation paragraph of the Discussion section:

“Previous research reveals a gradient pattern in which low-level representation synchrony occurs in primary cortical regions, whereas high-level representation synchrony occurs at hierarchically higher cortical regions such as the left IFG (43). The current study does not disentangle synchrony at different levels. Future studies may examine this issue using different paradigms.”

- 1- Stolk, A., Verhagen, L., & Toni, I. (2016). Conceptual alignment: How brains achieve mutual understanding. *Trends in cognitive sciences*, 20(3), 180-191.